# Overexpression of miR-210-3p Impairs Extravillous Trophoblast Functions Associated with Uterine Spiral Artery Remodeling

**DOI:** 10.3390/ijms22083961

**Published:** 2021-04-12

**Authors:** Heyam Hayder, Guodong Fu, Lubna Nadeem, Jacob A. O’Brien, Stephen J. Lye, Chun Peng

**Affiliations:** 1Department of Biology, York University, Toronto, ON M3J 1P3, Canada; heyam23@yorku.ca (H.H.); nadeem@lunenfeld.ca (L.N.); jaobr@my.yorku.ca (J.A.O.); 2Research Centre for Women’s and Infants’ Health, Lunenfeld-Tanenbaum Research Institute, Mount Sinai Hospital, Sinai Health System, Toronto, ON M5G 1X5, Canada; lye@lunenfeld.ca; 3Department of Obstetrics and Gynaecology, Faculty of Medicine, University of Toronto, Toronto, ON M5G 1E2, Canada; 4Department of Physiology, Faculty of Medicine, University of Toronto, Toronto, ON M5S 1A8, Canada; 5Centre for Research on Biomolecular Interactions, York University, Toronto, ON M3J 1P3, Canada

**Keywords:** placenta, preeclampsia, trophoblast, miR-210-3p, CDX2, cytokines, interleukin-1B, interleukin-8, CXCL1

## Abstract

Hsa-miR-210-3p has been reported to be upregulated in preeclampsia (PE); however, the functions of miR-210-3p in placental development are not fully understood, and, consequently, miR-210-3p’s role in the pathogenesis of PE is still under investigation. In this study, we found that overexpression of miR-210-3p reduced trophoblast migration and invasion, extravillous trophoblast (EVT) outgrowth in first trimester explants, expression of endovascular trophoblast (enEVT) markers and the ability of trophoblast to form endothelial-like networks. In addition, miR-210-3p overexpression significantly downregulated the mRNA levels of interleukin-1B and -8, as well as CXC motif ligand 1. These cytokines have been suggested to play a role in EVT invasion and the recruitment of immune cells to the spiral artery remodeling sites. We also showed that caudal-related homeobox transcription factor 2 (CDX2) is targeted by miR-210-3p and that CDX2 downregulation mimicked the observed effects of miR-210-3p upregulation in trophoblasts. These findings suggest that miR-210-3p may play a role in regulating events associated with enEVT functions and its overexpression could impair spiral artery remodeling, thereby contributing to PE.

## 1. Introduction:

Early placental development is a highly regulated process as the trophectoderm layer differentiates into multiple trophoblast lineages to form a branching network of villi [1]. Cytotrophoblast progenitor cells differentiate into two general pathways; they can either fuse to form a multinucleated monolayer of syncytiotrophoblasts or differentiate into the invasive extravillous trophoblasts (EVT) [2]. One important role of these invasive EVT is the remodeling of the maternal uterine spiral arteries. During the first trimester, EVTs form aggregates (plugs) to limit the maternal blood flow into the intervillous space (IVS) [3]. Toward the end of the first trimester and throughout the second trimester, EVTs that are in contact with the spiral arteries differentiate into endovascular EVTs (enEVTs) which replace the endothelial cells on spiral arteries and transform them into low resistance, large diameter vessels to establish low velocity uteroplacental perfusion [3,4]. Reduced trophoblast invasion and abnormal vessel remodeling can lead to prolonged placental oxidative stress, thus contributing to trophoblast dysfunction which is a hallmark of preeclampsia (PE) [4,5].

MicroRNAs (miRNA) are small non-coding, single-stranded RNA that regulate gene expression at the post-transcriptional and to a lesser extent transcriptional levels [1]. They have been shown to play a vital role in many biological processes including placenta development [1]. Considering the importance of miRNA in placenta development, many studies have investigated the effect of oxygen on miRNA expression within the placenta [1,5]. A group of miRNAs found to be upregulated by hypoxia, called hypoxamirs, were directly regulated by hypoxia-induced transcription factors [1]. Hsa-miR-210-3p is one of the most well-established hypoxamirs, and it was shown to be upregulated in disparate low oxygen environments such as cancer [6,7], myocardial infarction [8], ischemia [9] and PE [10,11]. miR-210-3p is directly induced by hypoxia-inducible factor 1 alpha (HIF1A) binding to the hypoxia response element (HRE) within the miR-210-3p promoter [10]. miR-210-3p can also be induced by another hypoxia regulated transcription factor, nuclear factor kappa-B subunit p50 (NFKB1) [12]. In addition, many studies have shown that miR-210-3p level was higher in placenta and serum from women with PE [4,11,12,13,14]. miR-210-3p plays a role in regulating cell proliferation, migration and invasion, cell cycle regulation, apoptosis, angiogenesis and mitochondrial function [7,8,15,16]. In trophoblasts, miR-210-3p was reported to decrease cell invasion by targeting various genes, such as iron-sulfur cluster assembly enzyme (ISCU) [17], thrombospondin type I domain containing 7A (THSD7A) [18], potassium channel modulatory factor 1 (KCMF1) [19] and fibroblast growth factor 1 (FGF1) [20]. miR-210-3p is also involved in the regulation of syncytialization [21] and mitochondrial respiration [14] in trophoblasts.

Caudal-related homeobox transcription factor 2 (CDX2) is a transcription factor known to be essential for the establishment of the trophoblast lineage in the late morula stage [22,23,24]. Studies have shown that overexpression of CDX2 in the EVT cell line, HTR8/SVneo, enhances cell invasion through activation of matrix metalloproteinase-9 (MMP9) and inhibition of MMP9 inhibitor, tissue inhibitor metalloproteinase-1 (TIMP1) [25] and that CDX2 expression is regulated by the PI3K/AKT pathway [25]. Additionally, CDX2 is regulated by Yes-associated protein (YAP), which has been shown to promote HTR8/SVneo invasion and reduce apoptosis [26].

Uterine spiral artery remodeling is an important part of proper placenta development. It involves a dynamic interaction among at least the decidual immune cells, EVT, endothelial and smooth muscle cells of the artery [27]. Many autocrine and paracrine factors secreted by EVTs, including cytokines, chemokines and growth factors, are known to play a role in the recruitment of immune cells to the site of remodeling [28,29]. Such cytokines and chemokines include interleukin-1β (IL1B), interleukin-8 (CXCL8) and CXC motif ligand 1 (CXCL1) [27,30,31,32]. IL1B, a pro-inflammatory cytokine, has been shown to promote EVT migration [33] and invasion [34]. CXCL8, a chemokine that is able to induce chemotaxis, has been shown to activate endothelial cell retraction and gap formation between adjacent cells [35]. CXCL1, also a chemokine, is secreted by trophoblasts and has been shown to play a role in the recruitment of regulatory T-cells needed for maternal tolerance of the fetus [36], as well as having an angiogenic role [28]. CXCL1 expression was also shown to be induced by IL1B and to promote trophoblast invasion [37].

The role of miR-210-3p in spiral artery remodeling is largely unknown. In this study, we examined the role of miR-210-3p in EVT functions associated with spiral artery remodeling and identified CDX2 as a novel target of miR-210-3p. We report here that overexpression of miR-210-3p or silencing of CDX2 inhibited: (1) migration and invasion of HTR8/SVneo cells; (2) outgrowth of first trimester placental villous explants; and (3) the potential of EVT to form endothelial-like networks. In addition, overexpression of miR-210-3p or knockdown of CDX2 decreased the mRNA level of ITGA1, PECAM1, CDH5, IL1B, CXCL8 and CXCL1, which are markers of enEVTs or known to promote EVT invasion and/or spiral artery remodeling. Our findings suggest that overexpression of miR-210-3p could lead to the impaired maternal spiral artery remodeling, thereby contributing to the pathogenesis of PE.

## 2. Results

### 2.1. miR-210-3p Level Decreases with Increasing Gestation in Healthy Pregnancies and Is Upregulated in PE

To evaluate the endogenous level of miR-210-3p across gestation, we quantified its expression in placental samples from healthy pregnancy ranging from 5 to 40 weeks, collected from a Canadian cohort using qRT-PCR (Figure 1A). The level of miR-210-3p was significantly lower during 26–40 weeks compared to 5–12 weeks of gestation. In addition, we measured miR-210-3p levels in placental samples from pre-term (26–36 weeks) and term (37–40 weeks) PE pregnancies and age-matched healthy controls. There was no significant difference in miR-210-3p levels between pre-term control and PE samples (Figure 1B). However, a significant increase in miR-210-3p levels in PE compared to the age-matched control pregnancies was observed in term samples (Figure 1C).

### 2.2. miR-210-3p Downregulates Migration and Invasion of HTR8/SVneo Cells and EVT Outgrowth in First Trimester Placental Explants

To further confirm the role of miR-210-3p in trophoblast motility, Transwell migration and invasion assays were conducted. Overexpression of miR-210-3p in HTR8/SVneo cells reduced their ability to migrate and invade (Figure 2A,C) while transfection with anti-miR-210-3p promoted cell migration and invasion (Figure 2B,D). Using the first trimester placental explant model, the role of miR-210-3p in EVT outgrowth was also investigated. In explants overexpressing miR-210-3p, EVT outgrowth was significantly impeded (Figure 2E). On the other hand, anti-miR-210-3p significantly promoted EVT outgrowth (Figure 2F).

### 2.3. miR-210-3p Reduced the Ability of HTR8/SVneo to Form Endothelial-Like Networks

To determine if miR-210-3p regulates the ability of trophoblasts to acquire angiogenic potentials, tube formation assays were performed. Overexpression of miR-210-3p in HTR8/SVneo cells led to a significant decrease in the total tube length and the number of branching points at both 24 and 48 h (Figure 3A). On the other hand, transfecting the cells with anti-miR-210-3p significantly increased the number of branching points at 48 h but had no significant effect on total tube length at both time points tested (Figure 3B).

### 2.4. Overexpressing miR-210-3p in HTR8/SVneo Decreased the mRNA Levels of Invasion and enEVT Differentiation Markers Especially IL1B, CXCL8 and CXCL1

To investigate the effect of miR-210-3p overexpression on known EVT and enEVT markers, we used qRT-PCR to measure the mRNA levels of these genes. HTR8/SVneo cells transfected with miR-210-3p mimics (Figure 4A) showed a significant decrease in key markers, ITGA1 and PECAM1, but no significant change in CDH5 and HLA-G was observed. Moreover, miR-210-3p also significantly decreased mRNA levels of the pro-inflammatory cytokine IL1B and chemokines CXCL8 and CXCL1, all of which have been shown to be important for trophoblast invasion and in the recruitment of immune cells to remodeling spiral arteries [27,34,37,38]. However, downregulation of endogenous miR-210-3p by anti-miR-210-3p showed no significant effects on the mRNA level of all of these markers (Figure 4B).

### 2.5. Upregulation of Endogenous miR-210-3p by Low Oxygen Tension Led to a Decrease in IL1B, CXCL8 and CXCL1 at 1% O_2_

It is well-established that miR-210-3p is upregulated in response to low-oxygen tension. We investigated if upregulating endogenous miR-210-3p by culturing HTR8/SVneo under 8%, 3% and 1% O_2_ can lead to change in the mRNA levels of cytokine/chemokine IL1B, CXCL8 and CXCL1. As expected, miR-210-3p was significantly upregulated in cells cultured at 1% O_2_ at both 24 (Figure 5A) and 48 h (Figure 5B). In addition, the mRNA levels of IL1B, CXCL8 and CXCL1 were all significantly downregulated at 1% O_2_ (Figure 5A,B).

### 2.6. CDX2 Is a Novel Target of miR-210-3p

Several genes have been reported to be targets of miR-210-3p and implicated in regulating trophoblast function [1,5,19,20]. In search of novel miR-210-3p targets, we conducted an in-silico search which showed that CDX2 mRNA contained a predicted miR-210-3p target site in the 3′ UTR. To confirm whether CDX2 is indeed targeted by miR-210-3p, we generated a luciferase reporter with the region of the 3′ UTR of CDX2 containing the predicted target site and a mutant construct with mutations in the seed region of the predicted binding site. Luciferase assays showed a decrease in activity after transfecting miR-210-3p mimic and this decrease was abolished when the predicted miR-210-3p binding site was mutated (Figure 6A). In addition, miR-210-3p decreased both CDX2 mRNA (Figure 6B) and protein (Figure 6C) levels.

### 2.7. CDX2 Knockdown Mimics the Effect of miR-210-3p Overexpression and Led to Decreased HTR8/SVneo Migration and Invasion and Decreased EVT Outgrowth in First Trimester Placental Explants

To investigate the role of CDX2 in trophoblasts, we designed a small interfering RNA (siRNA) molecule targeting CDX2. In Transwell migration assays, knockdown of CDX2 by siCDX2 significantly decreased cell migration (Figure 7A). Similarly, in Transwell invasion assay, siCDX2 decreased the number of invaded cells (Figure 7B). Using the first trimester explant model, we also investigated the role of CDX2 in EVT outgrowth. Treatment of the explants with siCDX2 led to a decrease in EVT outgrowth, measured at both 48 and 72 h (Figure 7C).

### 2.8. CDX2 Downregulation Reduced the Ability of HTR8/SVneo to Form Endothelial-Like Networks

To test whether CDX2 affects the angiogenic potential of trophoblasts, HTR8/SVneo cells were transfected with siCDX2 and tube formation assays were performed. Similar to the overexpression of miR-210-3p, knockdown of CDX2 using siRNA in HTR8/SVneo resulted in a significant decrease in the total tube length and the number of branching points at both 24 and 48 h (Figure 8A). Interestingly, co-transfecting siCDX2 and anti-miR-210-3p led to a partial rescue of the phenotype seen with siCDX2 alone at 24 h (Figure 8B).

### 2.9. CDX2 Knockdown in HTR8/SVneo Decreased the Levels of EVT and enEVT Differentiation Markers, As Well As IL1B, CXCL8 and CXCL1

To determine whether CDX2 affects multiple EVT and enEVT differentiation markers, HTR8/SVneo cells were transfected with siCDX2. Downregulation of CDX2 led to a significant decrease in the mRNA levels of ITGA1, PECAM1, CDH5, HLA-G, IL1B, CXCL8 and CXCL1 (Figure 9), thus mimicking the effects observed with miR-210-3p overexpression.

## 3. Discussion

MicroRNA have been shown to play an important role in every step of a successful pregnancy including trophectoderm development, implantation and placentation [1]. Cytotrophoblast differentiation towards both the syncytial and extravillous pathways have been shown to be regulated by multiple miRNAs [1,2,5,39,40,41]. Previous studies have reported that miR-210-3p inhibits trophoblast invasion [4,18,19]. In this study, we showed that miR-210-3p also inhibits the outgrowth of placental explants, the expression of enEVT markers, the ability of trophoblast to form endothelial-like networks and the mRNA levels of several cytokine/chemokines known to be involved in EVT invasion and spiral artery remodeling. These novel findings suggest that miR-210-3p plays a role in regulating uterine spiral artery remodeling.

During early placental development, some cytotrophoblasts differentiate into EVT that invade the decidua. A subset of EVT further differentiates into enEVT which acquires endothelium-like properties and replaces the endothelium in uterine spiral arteries during vascular transformation. In this study, we provided several lines of evidence to suggest that miR-210-3p exerts inhibitory effects on the acquisition of an enEVT-like phenotype. First, overexpression of miR-210-3p decreased cell migration and invasion, as well as first trimester placental explant outgrowth, while inhibition of endogenous miR-210-3p promoted cell migration/invasion and explant outgrowth. Second, miR-210-3p overexpression in HTR8/SVneo cells inhibited the ability of these cells to form endothelial-like networks. Third, miR-210-3p overexpression also decreased the mRNA levels of the invasive EVT and enEVT markers ITGA1 [2,40], CDH5 [42], PECAM1 [42], IL1B, CXCL8 and CXCL1 [2,40,43]. Although anti-miR-210-3p was able to increase cell migration, invasion and EVT outgrowth, it only significantly increased the branching points at 48 h and did not significantly increase enEVT marker mRNA levels. Since many factors are involved in promoting spiral artery remodeling [44,45], it is likely that partial inhibition of endogenous miR-210-3p alone is insufficient to increase angiogenic potential and enEVT marker expression.

Uterine decidual natural killer cell (dNK) and macrophage recruitment to the spiral arteries is required to initiate the remodeling process [2,27]. EVT has been shown to secrete different cytokines and chemokines that recruit these maternal immune cells into the spiral arteries [27,31,37,46,47]. Recently, we demonstrated that miR-218-5p promotes spiral artery remodeling by inducing enEVT differentiation and increasing cytokine/chemokine expression, including IL1B, CXCL8 and CXCL1 [40]. In the present study, we found that overexpression of miR-210-3p downregulated the mRNA level of IL1B, CXCL8 and CXCL1 in HTR8/SVneo cells. IL1B was shown to promote EVT migration [33], and it has been suggested to be secreted by trophoblast to regulate their own invasion into the decidua [46]. It was also shown that CXCL8 promotes trophoblast invasion [38] and that it is secreted by enEVT to stimulate endothelial cells secretion of CCL14 and CXCL6 which attract and recruit dNK cells and macrophages from the surrounding decidual tissue into the spiral arteries to accelerate the remodeling process [27]. CXCL1 has also been suggested to play a role in trophoblast-decidual communication and to act as a chemoattractant for neutrophils and leukocytes [46,48]. Inhibition of these cytokine and chemokine expressions further suggests that miR-210-3p regulates events associated with spiral artery remodeling.

CDX2 is a major transcription factor important for trophectoderm lineage and maintenance of trophoblast self-renewal [49,50,51,52]. CDX2 mRNA has been detected in multiple trophoblast cell lines such as JEG-3, TCL-1 [53] and HTR8/SVneo [25]; however, others reported that its protein level was undetectable in placental samples beyond 15 weeks of gestation [53,54]. Although a specific role of CDX2 in early differentiation of EVT pathways has not been reported, a recent study showed that CDX2 increased HTR8/SVneo motility [25]. In non-human species, CDX2 was shown to regulate multiple trophoblast genes [55], such as HAND1 and SOX15, both of which were shown to promote trophoblast differentiation into trophoblast giant cells in mouse [56,57]. In this study, using luciferase reporter assays, qRT-PCR and Western blot analyses, we demonstrated that CDX2 is a target of miR-210-3p and exerts promoting effects on enEVT functions. Specifically, we found that silencing of CDX2 mimicked the effect of miR-210-3p overexpression in inhibiting migration and invasion, first trimester EVT outgrowth, formation of endothelium-like networks, expression of invasive EVT and enEVT markers and expression of cytokine/chemokines. In addition, downregulating endogenous miR-210-3p partially attenuated the effect of siCDX2 on endothelial-like formation. These results suggest that CDX2 promotes enEVT functions associated with successful spiral artery remodeling. However, further experiments are needed to elucidate the mechanisms of how CDX2 regulates these processes.

We also investigated placental miR-210-3p levels across gestation in normal pregnancies and found that miR-210-3p levels are highest in the first trimester (Weeks 5–12) and are lowest in the third trimester (Weeks 26–40). During the first trimester, uterine spiral artery openings into the placental intervillous space (IVS) are obstructed by EVT plugs to protect the placenta and developing fetus from damage caused by oxidative stress. Instead, the IVS is filled with plasma and endometrial gland secretions to provide the necessary nutrients and physiological oxygen concentrations of around 2.5% prior to 10 weeks of gestation [3]. The observed high level of miR-210-3p in first trimester samples suggest a role of miR-210-3p in early placentation events where trophoblasts have reduced mitochondrial function and rely heavily on glycolysis for ATP production [3,58]. In cancer studies, miR-210-3p was shown to upregulate glycolysis [59], and, in trophoblasts, miR-210-3p targets ISCU, thus reducing mitochondrial function [14].

Failure to remodel uterine spiral arteries into low resistance vessels leads to villous shear damage and placental malperfusion, causing oxidative stress in the placenta [3,60]. This is a major mechanism underlying the development of PE, especially the severe, early-onset PE [3]. Using placenta samples from healthy and PE pregnancies, we observed higher miR-210-3p levels in placentas of PE women who delivered a baby at term. However, in women who delivered at preterm, we did not see a significant difference in miR-210-3p levels between PE patients and control subjects. Notably, the preterm control samples were all from women who entered the laboring process while most of the preterm PE women underwent cesarean section without labor. It is possible that the laboring process could affect miR-210 levels. It has been reported that preterm labor altered the ratio of miR-210 in the amnion and chorion [61]. Labor is also associated with increased NFKB1 activity in the placenta [62,63,64,65,66], and NFKB1 is known to induce miR-210-3p levels [12,67]. The majority of studies that reported an upregulation of miR-210-3p in PE only used term samples [18,19,68,69]. Interestingly, several studies have shown that miR-210-3p levels in the serum of women who developed PE later in gestation were higher during the second trimester than in women who did not develop PE [69,70]. Furthermore, it has been reported that miRNAs can be secreted from trophoblasts and enter maternal circulation [71,72]. This raises the intriguing possibility that increased levels of circulating miR-210-3p in the serum of women with PE may be secreted from oxidative-stress damaged trophoblasts. The secretion of excess miR-210-3p from these trophoblasts may explain the lack of detectable intracellular upregulation of miR-210-3p in our preterm PE samples. As gestation continues, accumulation of oxidative damage may increase intracellular levels above what can be efficiently secreted, thus leading to the observed increase of intracellular miR-210-3p in term samples. As circulating miRNAs can be taken up by target cells [1], it is possible that enEVT within remodeling spiral arteries may be exposed to increased levels of circulating miR-210-3p, inhibiting functions associated with successful spiral artery remodeling and contributing to a potential positive feedback of PE pathogenesis. This possibility requires further investigation.

In summary, we used multiple approaches to investigate the expression and function of miR-210-3p in human placenta. We confirmed previous findings that miR-210-3p level is upregulated in term PE. In addition, overexpression of miR-210-3p decreased trophoblast migration, invasion, placental explant outgrowth, the ability of trophoblasts to form endothelial-like network and the expression of enEVT markers and cytokine/chemokines known to be involved in trophoblast/immune cell communication during spiral artery remodeling. We also identified CDX2 as a novel target of miR-210-3p. Our findings suggest that the miR-210-3p plays a role in constraining enEVT functions associated with successful spiral artery remodeling, in part by downregulating CDX2. Upregulation of miR-210-3p in EVT may contribute to the impaired maternal spiral artery remodeling observed in PE. However, there are several limitations to our study. For example, we only measured mRNA levels of marker gene expression, and, although we showed that miR-210-3p reduced the ability of trophoblasts to form network-like structures, more studies are required to confirm the role of miR-210-3p in enEVT differentiation and spiral artery remodeling in human pregnancy.

## 4. Material and Methods

### 4.1. Clinical Samples and Tissue Collection

The human placental tissue samples were collected through the BioBank program at the Research Centre for Women’s and Infants’ Health at Mount Sinai Hospital (MSH) (Toronto, ON, Canada) and approved by the MSH Research Ethics Board, including first trimester (5–12 weeks, *n* = 8), second trimester (13–25 weeks, *n* = 10), pre-term (26–36 weeks, *n* = 13), and term (37–40 weeks, *n* = 17) samples. First and second trimester samples were collected with informed consent from healthy patients undergoing elective termination of pregnancy at the Morgentaler Clinic (Toronto, ON, Canada). Placental samples from patients with HIV, hepatitis, or missed miscarriage were excluded. Pre-term samples were collected by caesarean section following laboring after fetal distress or spontaneous premature rupture of the membranes. All pre-term samples were examined by a placental pathologist and only placenta without gross abnormalities or signs of chorioamnionitis were included. Third trimester placental samples were collected from only appropriate for gestational age vaginal or caesarean section deliveries. Pre-eclamptic samples were collected from both pre-term (*n* = 14) and term (*n* = 4) pregnancies. Healthy and PE donors’ clinical data are summarized in Table 1.

### 4.2. Cell Lines and Cell Culture

Experiments were conducted using an immortalized human first trimester trophoblast cell line, HTR8/SVneo, kindly provided by Dr. Charles Graham (Queen’s University, Kingston, ON, Canada) and were maintained as previously described [40,73]. Briefly, HTR8/SVneo cells were cultured in HyClone™ classical liquid media RPMI 1640 with L-glutamine (GE Healthcare Life Sciences, Ottawa, ON, Canada) supplemented with 10% fetal bovine serum (FBS) (GIBCO^®^ Life Technologies, Mississauga, ON, Canada) and grown in humidified environment of 5% CO_2_ and 37 °C. For oxygen tension experiments, cells were incubated in Sanyo tri-Gas incubators set at 21%, 8%, 3% or 1% O_2_ levels and grown in humidified environment of 5% CO_2_ and 37 °C.

### 4.3. Transient Transfections and Treatments

Transient transfection of miR-210-3p mimics or inhibitor or small interfering RNA oligomers (100 nM) (Table 2) was conducted using Lipofectamine RNAiMAX^®^ reagent (Invitrogen, Life Technologies, Mississauga, ON, Canada). A modified protocol was used to optimize transfection efficiency and cell survival using a Lipofectamine volume of 2.4 µL per well in OMEM medium [40,73]. Cells were transfected 24 h post-seeding for 5 h in 6-well plates seeded at 6.0 × 10^4^ cells per well. Transfection mix was removed, and cells were recovered with serum containing media for 24 h after transfection before proceeding to functional assays and for 48 h for RNA or Western blot analysis.

### 4.4. RNA Extraction, Reverse Transcription and Quantitative Real Time PCR (qRT-PCR)

Total RNA was extracted from cells or tissues using RiboZol™ RNA Extraction Reagent (VWR Life Science, Mississauga, ON, Canada) following manufacturer’s protocol with some modifications. Namely, during the RNA precipitation step, instead of the recommended 10 min at room temperature in isopropanol, 30 min at −20 °C was performed. RNA concentration was measured using NanoDrop 2000 spectrophotometer (Thermo Scientific, Mississauga, ON, Canada). NCode™ miRNA first-strand cDNA synthesis kit (Invitrogen, Life Technologies, Mississauga, ON, Canada) was used to detect miR-210-3p or internal control U48 in clinical samples following manufacturer’s protocol. For measuring has-miR-210-3p or U6 post-transfection in cells, TaqMan^®^ miRNA assay (Life Technologies, Mississauga, ON, Canada) was used following manufacturer’s protocol.

To measure the levels of mRNA, 1.0 μg of total RNA was reversed transcribed into cDNA using M-MuLV Reverse Transcriptase (New England Biolabs), following the manufacturer’s protocol. qRT-PCR was performed using EvaGreen qPCR 2X Master mix (Applied Biological Materials Inc., Richmond, BC, Canada) following the manufacturer’s recommendation and samples were normalized to CYC1 internal control [40,73] using the 2^−ΔΔCT^ method. Primers used were designed using NCBI primer blast and are listed in Table 3. Some of these primers have been reported previously [40].

### 4.5. Transwell Migration and Invasion Assay

Transwell migration and invasion assays were conducted using polycarbonate membrane 24-well Transwell inserts with 8 µm pore size (Costar, Corning™ Inc., Ottawa, ON, Canada). However, in the invasion assay, the transwell inserts were coated with 100 μL of Cultrex PathClear^®^ growth factor reduced, phenol red-free Matrigel (Trevigen Inc., Burlington, ON, Canada) diluted with serum-free RPMI 1640 medium into a final concentration of 0.15 mg/mL. Cells were transfected with NC, siRNA, miR-210-3p, or anti-miR-210-3p 24 h prior to migration and invasion assays. Cells were then collected using Accutase (Corning Inc.) and seeded on the top of the transwell insert at a density of 1.5 × 10^4^ cells for migration and 2.0 × 10^4^ cells for invasion per filter. Cells were seeded into the insert in serum free media while 10% FBS containing media was added outside the Transwell.

At 18 h post seeding into the transwell, cells were fixed and stained with Harleco Hemacolor Staining Kit (EMD Millipore, Oakville, ON, Canada). Non-migrated/invaded cells on the top of the membranes were wiped off using a cotton swab and membranes were cut with a blade and mounted on slides for quantification. Cells were visualized and photographed using a dissection microscope (Leica Microsystems, at 1.25×) and the numbers of cells migrated/invaded were counted using ImageJ as described previously [74].

### 4.6. First-Trimester Human Placenta Explant Culture

Human placenta explant culture was performed as described previously [40,75,76,77]. Briefly, first trimester placentas (6–9 weeks) from elective terminated pregnancies were obtained through the Research Center for Women’s and Infants’ Health BioBank Program and the Lunenfeld-Tanenbaum Research Institute (Toronto, ON, Canada) on the day of procedure. Placentas were carefully dissected and explants with potential EVT columns were placed onto 12 mm Transwell inserts (EMD Millipore, Oakville, ON, Canada) pre-coated with 200 µL of phenol red-free Matrigel (Corning™ Inc., Ottawa, ON, Canada) in a 24-well plate. Explants were then left overnight in a humidified environment of 3% O_2_, 5% CO_2_ and 37 °C before adding serum-free DMEM-F12 medium supplemented with 100 µg/mL Normacin™. After two days of culture, explants with successful EVT outgrowths were selected and randomly distributed for treatment with 200 nM oligomers. Explants were photographed immediately after adding the treatment and then at 24, 48 and 72 h using Leica DFC400 camera attached to a dissecting microscope. Images were analyzed for area of EVT outgrowth using ImageJ where the total growth area was measured at each time point, then the area at each subsequent time points for each explant was divided by its initial area at time 0 h. Before combining samples from different experiments/placentas, each experiment treatment was normalized to the average of the control group of that experiment.

### 4.7. Tube Formation Assay

Tube formation assay was conducted after 24 h recovery post-transfection (as mentioned above). Cells were stained with a 1 μM Cell Tracker™ Green CMFDA (Invitrogen, Life Technologies) in serum free media for 45 min and then collected using Accutase (Corning™ Inc., Ottawa, ON, Canada). Cells were then seeded at the density of 2.0 × 10^4^ cells per well into a 96-well plate pre-coated with 30 µL per well of undiluted growth factor reduced, phenol red-free Matrigel (Cultrex PathClear^®^ 15.4 mg/mL). Plate was put into the IncuCyte^®^ live-cell analysis system (Essen BioSience, Ann Arbor, MI, USA) for imaging at 4× for at least three days. Images were then acquired and analyzed in ImageJ using the NeuronJ plugin as previously described [78].

### 4.8. Protein Extraction and Immunoblot Analysis

After cell were transfected and allowed to recover for 48 h, cells were lysed with RIPA buffer (50 mM Tris/HCl, 150 mM NaCl, 1 mM EDTA, 1% Triton-X, 0.5% NP-40, 0.1% SDS, pH 7.4) supplemented with Pierce protease and phosphatase inhibitors (Thermo Scientific). Equal amounts of protein were separated by SDS-polyacrylamide gel electrophoresis and transferred to a polyvinylidene difluoride membrane (EMD Millipore) overnight at 4 °C. Membranes were blocked in blocking buffer (5% skim milk in 1× Tris-buffered Saline and Tween-20) for 1 h at room temperature. Membranes were then incubated overnight at 4 °C with the primary antibodies β-Actin (Cell Signaling 3700S) 1:1000, CDX2 (Cell Signaling 3977S) 1:500. Membranes were then washed with 1× TBST buffer and probed with horseradish peroxidase-conjugated secondary antibodies (Anti-mouse (EMD Millipore 12349MI) 1:4000, Anti-Rabbit (EMD Millipore 12348MI) 1:4000) at room temperature for 1.5 h. Signals were detected using Clarity™ Western ECL substrate kit (Bio-Rad Laboratories Ltd., Mississauga, ON, Canada).

### 4.9. Luciferase Reporter Assay

Luciferase reporter construct was generated by cloning a portion of the CDX2 3′ UTR containing the predicted miR-210-3p target sequence into pMIR-Report™ plasmid (Ambion) downstream of the luciferase gene. Briefly, the wild type CDX2 3′ UTR fragment from nucleotide 1252 to 2183 was amplified using HTR8/SVneo RNA as a template. Mutations at the seed region of the predicted miR-210-3p targeting site was introduced by PCR using primers with a substitution of four nucleotides (CACA to GTGT). Both wild type and the mutated fragments were cloned into the pMIR-Report plasmid using SpeI and PmeI restriction enzymes and finally sequenced to confirm their identities. Luciferase assay was performed using the Dual-Luciferase^®^ reporter assay system as per manufacturer’s protocol (Promega). HTR8/SVneo cells were transfected with 200 ng/mL of the Luciferase construct containing wild type or mutated CDX2 3′UTR, 25 ng/mL Renilla construct (pRL-TK, Promega) and either 20 nM of miR-210-3p or the scrambled control. At 24 h after transfection, cells were lysed and luciferase activity was measured.

### 4.10. Statistical Analysis

For comparison between more than two groups, one- or two-way ANOVA, depending on dataset, followed by Tukey’s multiple comparisons analysis was conducted using GraphPad Prism software (GraphPad Software Inc., San Diego, CA, USA). However, for comparison between two groups, un-paired Student’s t-test was used. A *p* < 0.05 was considered significant. To identify outliers in clinical samples, ROUT outliers test was performed (Q = 1.0%).

## Figures and Tables

**Figure 1 ijms-22-03961-f001:**
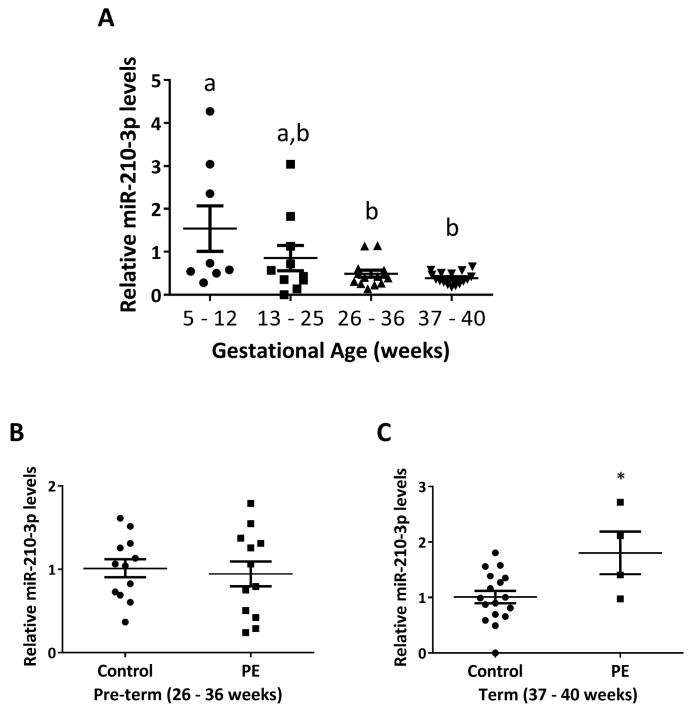
Expression levels of miR-210-3p in human placentas from healthy and PE pregnancies. (**A**) miR-210-3p expression across gestation. miR-210-3p levels was significantly lower in third trimester (Weeks 26–40) compared to first trimester (Weeks 5–12). Different letters above bars represent statistical significance. (**B**) miR-210-3p levels in pre-term (Weeks 26–36) healthy controls and PE placentas. There was no significant difference in miR-210-3p between the two groups. (**C**) Expression levels of miR-210-3p in healthy and PE term (Weeks 37–40) placentas. miR-210-3p levels were significantly higher in PE compared to healthy age-matched controls. Data represent mean ± SEM. * *p* < 0.05.

**Figure 2 ijms-22-03961-f002:**
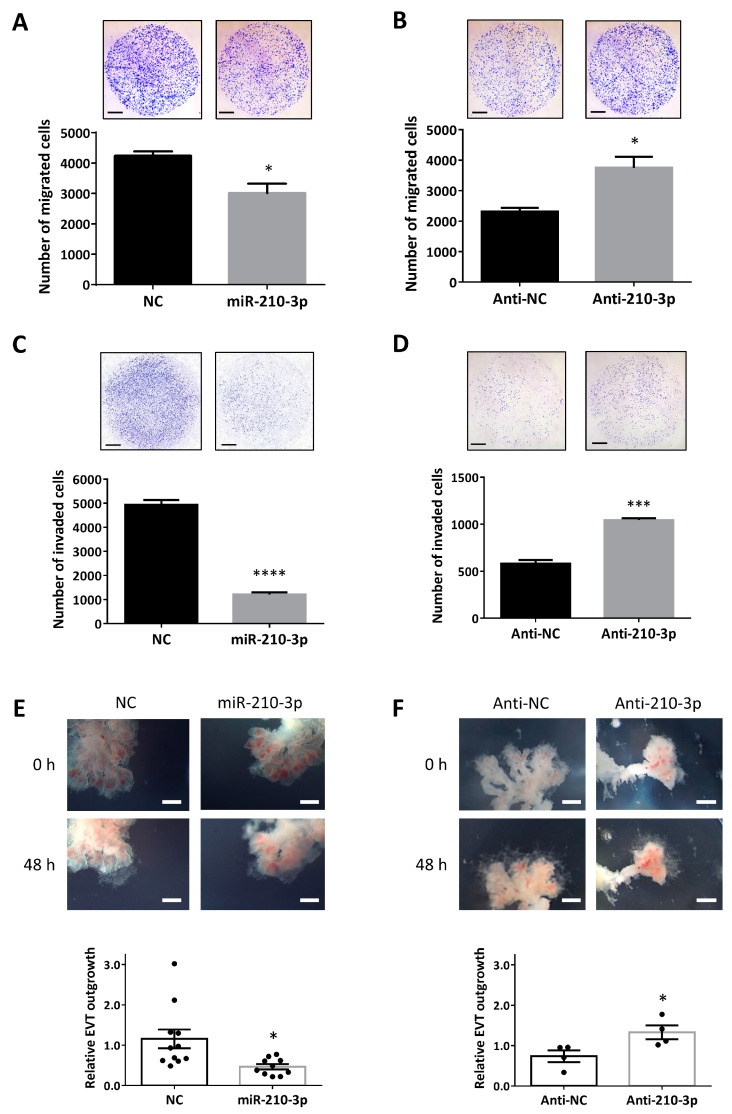
miR-210-3p inhibits trophoblast migration and invasion and EVT outgrowth. (**A**) Transwell migration assay using HTR8/SVneo cells transfected with miR-210-3p mimic (100 nM). Overexpression of miR-210-3p significantly reduced cell migration, when compared with cells transfected with a non-targeting control (NC, *n* = 3). (**B**) Transwell migration assay with cells transfected with anti-miR-210-3p (100 nM). Inhibition of endogenous miR-210-3p significantly increased the number of migrated cells compared to the negative control (anti-NC, *n* = 3). (**C**) Transwell invasion assay in cells overexpressing miR-210-3p. Trophoblast invasion ability was significantly decreased following miR-210-3p transfection (*n* = 5). (**D**) Invasion assay with HTR8/SVneo cells transfected with anti-miR-210-3p. The number of invaded cells was significantly increased after anti-miR-210-3p transfection compared to anti-NC (*n* = 3). (**E**) First-trimester (Weeks 6–9) placental explants were placed on Matrigel and treated with miR-210-3p or its control (200 nM) for 48 h. A significant decrease in EVT outgrowth in miR-210-3p-treated explants was observed (*n* = 11 NC, *n* = 10 miR-210-3p). (**F**) First-trimester placental explants treated with anti-miR-210-3p (200 nM) for 48 h had a significant increase in EVT outgrowth (*n* = 4 each group). Data represent mean ± SEM from representative experiments. * *p* < 0.05, *** *p* < 0.001, **** *p* < 0.0001. Scale bars: (**A**–**D**) 100 µm; and (**E**,**F**) 1 mm.

**Figure 3 ijms-22-03961-f003:**
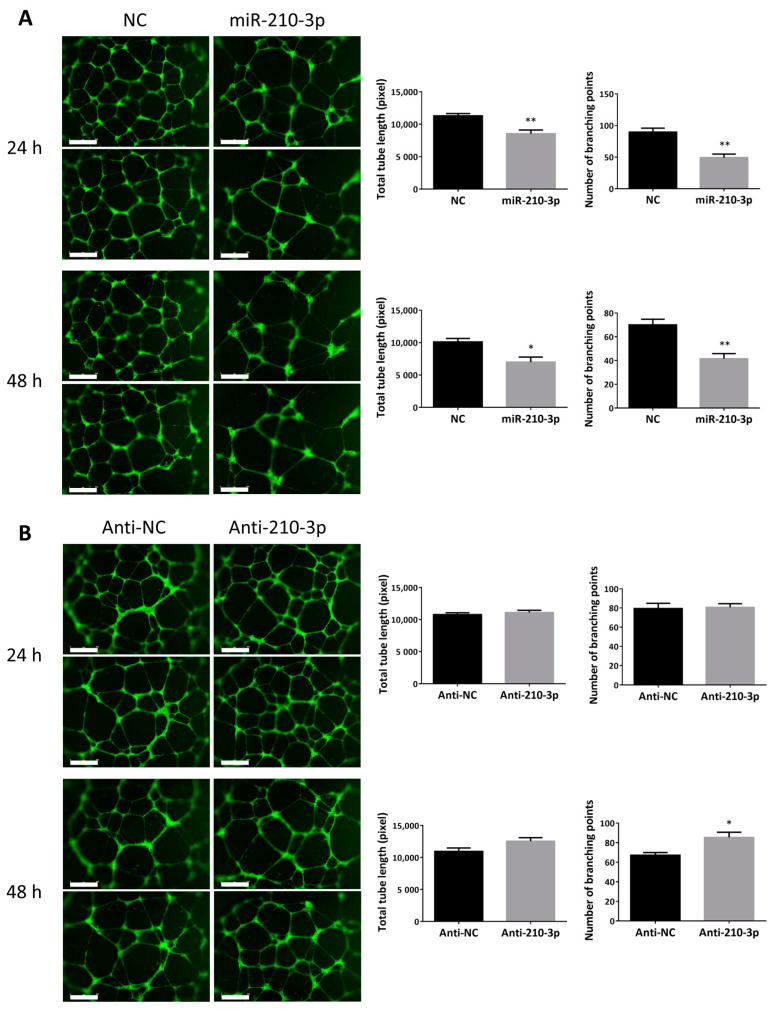
Overexpression of miR-210-3p reduces the ability of trophoblasts to form endothelial-like networks. (**A**) HTR8/SVneo cells transfected with 100 nM of miR-210-3p mimic showed a significant reduction in both the total length of the network formed and in the number of branching points at 24 and 48 h, as compared with cells transfected with non-targeting control (NC, *n* = 4). (**B**) Cells transfected with anti-miR-210-3p (100 nM) showed no change in the total tube length and the number of branching points at 24 h but showed a significant increase in the number of branching points at 48 h (*n* = 5) compared to the negative control (anti-NC). Data represent mean ± SEM from representative experiments. * *p* < 0.05, ** *p* < 0.01. Scale bar is 800 µm.

**Figure 4 ijms-22-03961-f004:**
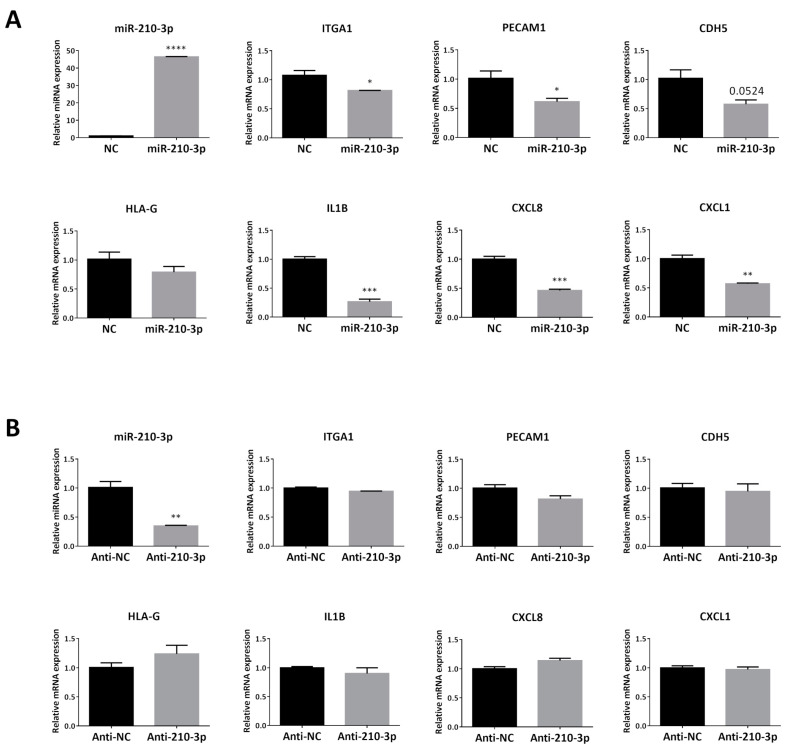
Overexpression of miR-210-3p downregulates mRNA levels of enEVT markers and cytokines. (**A**) In HTR8/SVneo cells transfected with 100 nM of miR-210-3p mimic for 48 h, mRNA levels of ITGA1, PECAM1, IL1B, CXCL8 and CXCL1 were significantly decreased compared with cells transfected with its non-targeting control (NC) while a decrease in CDH5 and HLA-G mRNA levels was not significant (*n* = 3). (**B**) Transfection of 100 nM anti-miR-210-3p for 48 h significantly decreased endogenous miR-210-3p levels compared to anti-NC miRNA inhibitor control but did not alter enEVT marker or cytokine mRNA levels (*n* = 3). Data represent mean ± SEM. * *p* < 0.05, ** *p* < 0.01, *** *p* < 0.001.

**Figure 5 ijms-22-03961-f005:**
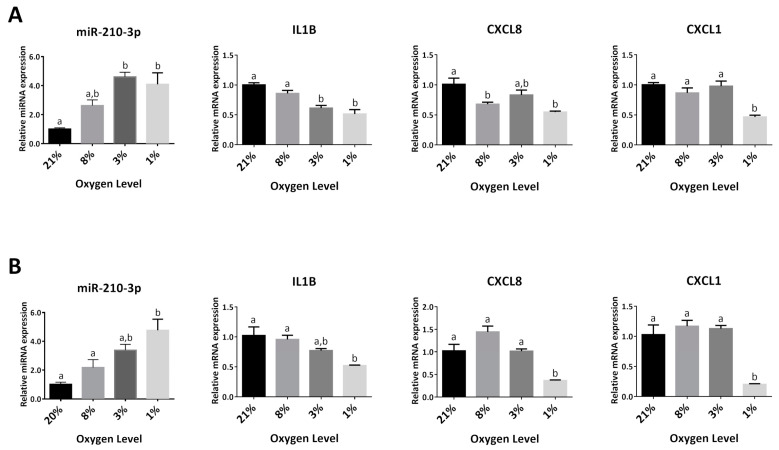
Low oxygen tension upregulates miR-210-3p and reduces cytokine levels. HTR8/SVneo cells cultured under different O_2_ levels for 24 h (**A**) or 48 h (**B**) (*n* = 3). miR-210-3p levels were upregulated while IL1B, CXCL8, and CXCL1 mRNA were downregulated by low O2 tension. Data represent mean ± SEM from representative experiments. Different letters above bars represent statistical significance. *p* < 0.05 was considered significant.

**Figure 6 ijms-22-03961-f006:**
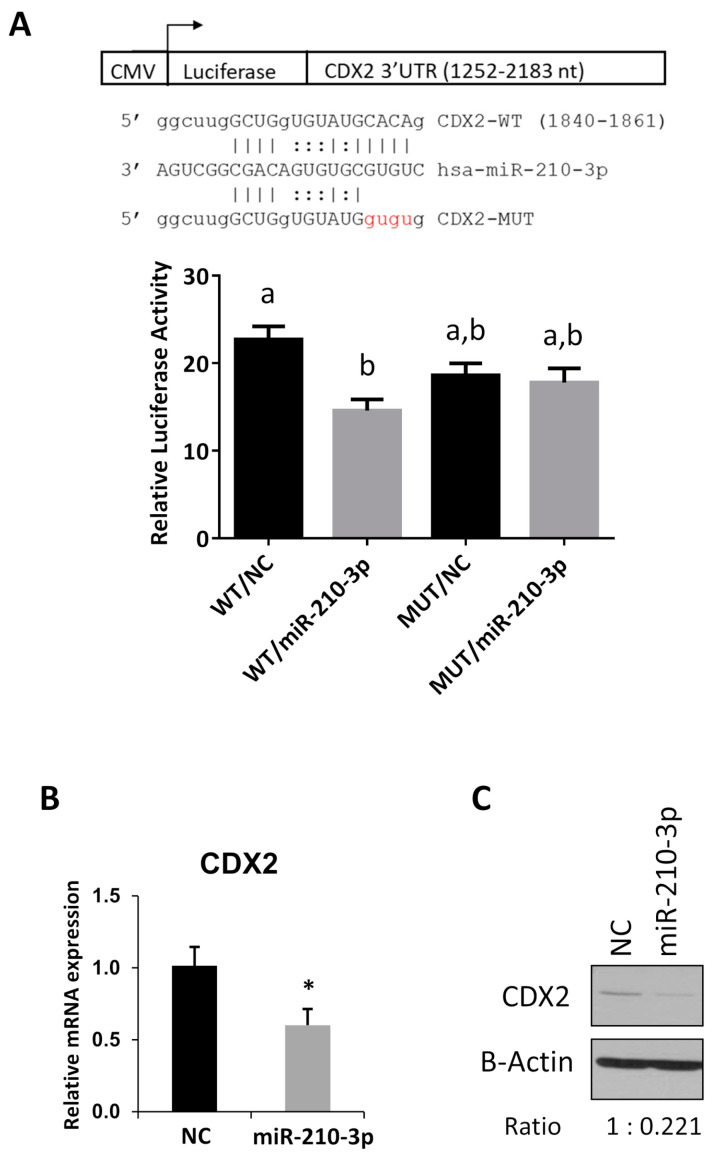
CDX2 is a novel target of miR-210-3p. (**A**) Luciferase reporter assay. The predicted miR-210-3p target site in the 3′ UTR of CDX2 was cloned into a luciferase reporter vector in its wild type (WT) sequence or after introducing mutations (MUT) to disrupt the predicted miR-210-3p binding site. Cells co-transfected with WT vector and miR-210-3p mimic decreased luciferase activity compared to cells transfected with the non-targeting control (NC). No reduction in luciferase activity by miR-210-3p was observed when the putative miR-210-3p binding site was mutated. Different letters above bars denote statistical significance (*p* < 0.05) (*n* = 5). (**B**) Overexpressing miR-210-3p led to a significant reduction in CDX2 mRNA level in HTR8/SVneo cells (*n* = 3). (**C**) CDX2 protein levels in cells transfected with miR-210-3p mimic was decreased compared to NC. Data represent mean ± SEM from representative experiments. * *p* < 0.05.

**Figure 7 ijms-22-03961-f007:**
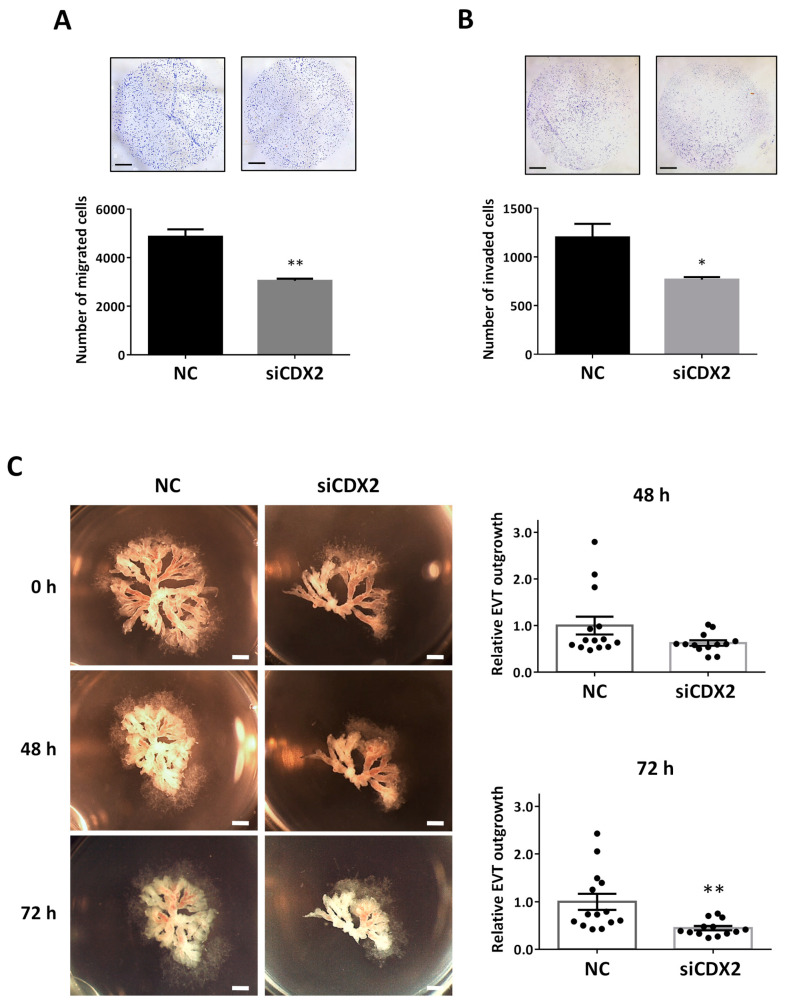
CDX2 promotes trophoblast migration, invasion and EVT outgrowth. (**A**) Transwell migration assay was conducted using HTR8/SVneo cells transfected with 100 nM of siCDX2. Downregulation of CDX2 led to a significant decrease in the number of migrated cells compared to the non-targeting control (NC) (*n* = 3). (**B**) In Transwell invasion assay, transfection with siCDX2 resulted in a significant decrease in the number of invaded cells (*n* = 3). (**C**) First-trimester (Weeks 6–9) placental explants treated with siCDX2 (200 nM) showed a significant decrease in the area of EVT outgrowth at 72 h, when compared with NC (*n* = 14 NC, *n* = 13 siCDX2). Data represent mean ± SEM from representative experiments. * *p* < 0.05, ** *p* < 0.01. Scale bars: (**A**,**B**) 100 µm; and (**C**) 1 mm.

**Figure 8 ijms-22-03961-f008:**
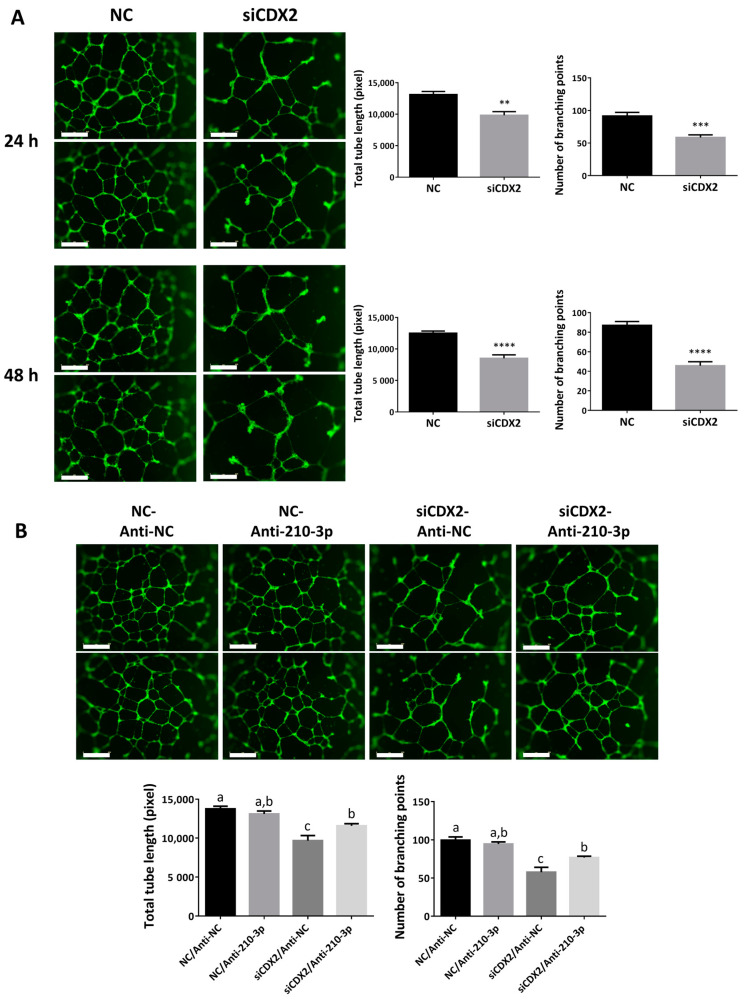
CDX2 promotes trophoblast potential to form endothelial-like networks. (**A**) Knockdown of CDX2 decreased total tube length and the number of branching points at both 24 and 48 h (*n* = 6). (**B**) The effect of downregulating CDX2 on trophoblast ability to form endothelial-like networks was partially reversed by downregulating endogenous miR-210-3p. HTR8/SVneo cells were transfected with 100 nM of siCDX2 or its non-targeting control (NC), along with anti-miR-210-3p or its negative control (anti-NC). siCDX2 decreased total tube length and the number of branching points but this effect was partially reversed by anti-miR-210 (*n* = 6) compared to those transfected with anti-NC. Data represent mean ± SEM from representative experiments. Deferent letters above bars denote statistical significance (*p* < 0.05). ** *p* < 0.01, *** *p* < 0.001, **** *p* < 0.0001. Scale bar is 800 µm.

**Figure 9 ijms-22-03961-f009:**
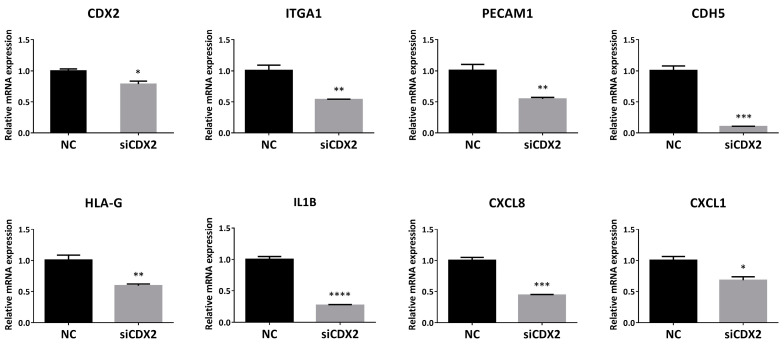
Silencing of CDX2 using siRNAs significantly inhibited mRNA levels of enEVT markers, as well as IL1B, CXCL8 and CXCL1. HTR8/SVneo cells were transfected with 100 nM of siCDX2 or the non-targeting control (NC) and RNA was extracted 48 h post-transfection (*n* = 3). Data represent mean ± SEM from representative experiments. * *p* < 0.05, ** *p* < 0.01, *** *p* < 0.001, **** *p* < 0.0001.

**Table 1 ijms-22-03961-t001:** Clinical information for control and age-matched PE patients. Data reported represent mean ± SEM.

	Pre-Term Control	Pre-Term PE	*p* Value	Term Control	Term PE	*p* Value
Gestational age (weeks)	29.83 ± 0.51	30.27 ± 0.36	0.4824	38.32 ± 0.14	37.25 ± 0.25	0.0030
Maternal age (years)	31.95 ± 1.15	30.8 ± 1.85	0.6087	33.36 ± 0.73	34.8 ± 2.21	0.4383
Systolic blood pressure (mmHg)	120.56 ± 2.9	165.13 ± 2.93	<0.0001	118.88 ± 2.11	166.8 ± 6.05	<0.0001
Diastolic blood pressure (mmHg)	73.83 ± 2.27	106.67 ± 1.9	<0.0001	77.04 ± 1.79	101.2 ± 1.46	<0.0001
Proteinuria (g/24 h)	None	3.46 ± 0.13	-	None	2.41 ± 0.25	-
% Laboring	100%	35.7%	0.0004	11.8%	75%	0.0276

**Table 2 ijms-22-03961-t002:** Oligomer sequences and reagents.

Oligomer Name	Sequence: 5′ → 3′
NC (non-targeting control) GenePharma (Shanghai, China)	Sense: UUCUCCGAACGUGUCACGUTT Anti-sense: ACGUGACACGUUCGGAGAATT
hsa-miR-210-3p GenePharma (Shanghai, China)	Sense: CUGUGCGUGUGACAGCGGCUGATT Anti-sense: UCAGCCGCUGUCACACGCACAGTT
siCDX2 GenePharma (Shanghai, China)	Sense: CCAGGACGAAAGACAAAUATT Anti-sense: UAUUUGUCUUUCGUCCUGGTT
Anti-NC RiboBio (Guangzhou, China)	RiboBio™ miRNA Inhibitor, Negative Control
Anti-hsa-miR-210-3p RiboBio (Guangzhou, China)	RiboBio™ anti- hsa-miR-210-3p Inhibitor

**Table 3 ijms-22-03961-t003:** Primer sequences used for qRT-PCR.

Primer Name	Sequence: 5′ → 3′
miR-210-3p	FP: GTGACAGCGGCTGAA RP: NCode Universal primer
U48	FP: CCCAGGTAACTCTGAGTGTGTC RP: NCode Universal primer
U6	FP: CGCAAGGATGACACGCAATT RP: NCode Universal primer
ITGA1	FP: GCTGGCTCCTCACTGTTGTT RP: CACCTCTCCCAACTGGACAC
PECAM1	FP: ATTGCAGTGGTTATCATCGGAGTG RP: CTCGTTGTTGGAGTTCAGAAGTGG
CDH5	FP: GCCAGTTCTTCCGAGTCACA RP: TTTCCTGTGGGGGTTCCAGT
HLA-G	FP: CTGACCCTGACCGAGACCTGG RP: GTCGCAGCCAATCATCCACTGGA
IL1B	FP: AATCTGTACCTGTCCTGCGTGTT RP: TGGGTAATTTTTGGGATCTACACTCT
CXCL8	FP: CAGAGACAGCAGAGCACACA RP: GGCAAAACTGCACCTTCACA
CXCL1	FP: CAGGGAATTCACCCCAAGAACA RP: GGATGCAGGATTGAGGCAAGC
CDX2	FP: CGCTTCTGGGCTGCTGCAAAC RP: CGACTGTAGTGAAACTCCTTCTCC
CYC1	FP: CAGATAGCCAAGGATGTGTG RP: CATCATCAACATCTTGAGCC

## Data Availability

Not applicable.

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
