# Peer review of "Overexpression of miR-210-3p Impairs Extravillous Trophoblast Functions Associated with Uterine Spiral Artery Remodeling"

_ijms, 2021, doi:10.3390/ijms22083961_

Round 1
Reviewer 1 Report
Hayder et al. evaluated the function of miR-210-3p in the human placenta. The study is well-executed, and several methods have been used to investigate their hypothesis. There are a few parts in the methodology and results which need some degree of clarifications:
1- The author concluded that the expression of miR-210=3p decreased with increasing gestational age. This is not correct as there is no significant difference in the expression level of this miRNA between 13-25wks and later stages. The only detected difference was between 5-12wks and >26. So it is not correct to conclude the expression decreased gradually (also line 330).
2- Result-2.1. The Term-PE group consists of only four samples, and their expressions are not homogenous. The data need to be tested for equal variances and also normality of distribution prior to statistical analysis. These might change the result of this section (Figure 1C). Is it possible to add more samples to this group?
3- Result-2.2. Why targeting NC (Anti-NC) has an effect on the cell migration number? (Figure 2 B and D). It is expected that targeting random oligomers doesn’t change the results (control). Proper controls that their expression doesn’t change the migration/invasion need to be used for this section.
4- All the figures are blurry.
5- Result-2.3. Figure 3. The bar indicating the size cannot be visualized. Are the figures in the same magnification?
6- Some of the figures are not mentioned in the text.
7- Result-2.4. What was used as transfection control? A scrambled (random) probe is needed for both overexpression and knocking-down experiments.
8- Result-2.5. Why did the author not include the result of normal (~5%) CO2 level?
9- Same as comment #5 for Figure 8.
10- Line 387: does AGA need to be capitalized?
11- Please provide a reference for the transfection method (4.3) as it is not the same as the manufacturer’s protocol.
12- Needs to provide references for the reference genes.
Author Response
We thank the reviewer for the insightful comments and suggestions.

Reviewer 2 Report
- This manuscript conducted many assays to confirm the functions of miR-210-3p on HTR-8/SVneo cell line and trophoblast villi. These assays were good and convincing. However, there is a logic shortage. As demonstrated in Discussion (line 279-283), these phenomena took place at the early stage of placenta development. The authors used HTR-8/SVneo to mimic the early development of placenta development. However, miR-201-3p was differentially expressed between control and PE at the late stage of gestation (Figure 1). In summary, the authors manipulated the un-varied miR-201-3p in the early stage of placenta development to alter the placenta functions in the early stage of placenta development. This is not convincing enough. Please address this issue in the discussion section or defense against it.
- In this study, the authors used qPCR to examine miR-210-3p in placenta tissue (Figure 1). Since placenta is a complicated tissue, composed of many types of cells. Please used IHC to confirmed the differential expression of miR-201-3p.
- The authors use miRNA mimic or inhibitor to transfect EVT tissue, followed by observing outgrowth (Figure 2E, F). Did the authors used qPCR to confirm the success of transfection. In other words, by how many times miR-210-3p was amplified or repressed?
Author Response
Thank you for your comments and suggestions. Please see our response in the attached file.

Reviewer 3 Report
This is a work that I read with great interest and enthusiasm. The paper is very well presented, well written, clear and provides all the necessary information in the introduction and methods. Because the role of the different miRNAs in preeclampsia is not fully understood, it is a valuable work in this regard, with adequate experiments to support the conclusions. However, I suggest that to complete it properly the authors should:
1) Define how they calculated the sample size for each experiment.
2) Define whether or not the women from whom the third trimester placenta samples from cesarean or vaginal births were obtained were in labor. If they had labor, discuss how this fact may interfere with the interpretation of the results.
3) Include the p values in the table describing the clinical characteristics of the patients to establish whether or not there are significant differences, especially in maternal age. In this case, explain whether this may be a factor in the interpretation of the results.
4) Include an analysis of strengths and weaknesses in the discussion.
Minor concerns:
1) Line 388 Pre-eclamptic samples collected from both pre- 388 term (n = 14) and term (n = 4). This sentence should be rewritten as: Pre-eclamptic samples were collected from both pre- 388 term (n = 14) and term (n = 4) pregnancies.
2) Line 421 manufacture's, add the missing r
3) Line 159 There was also a decrease, although not significant. This sentence should be rewritten because if it is not significant, there is no real decrease.
Author Response

(The authors gave the same response as above.)

Round 2
Reviewer 1 Report
Thanks for the edits. Please also remove/modify the sentence "In this study, we found that miR-210-3p levels in the placenta were negatively correlated with gestational age." from the abstract since no correlation analysis was performed in this study. Please use the same language as in the results and discussion regarding this miRNA expression during gestation.
Author Response
Comment: Please also remove/modify the sentence "In this study, we found that miR-210-3p levels in the placenta were negatively correlated with gestational age." from the abstract since no correlation analysis was performed in this study. Please use the same language as in the results and discussion regarding this miRNA expression during gestation.
Response: Thank you. The sentence has removed from the Abstract. We also checked the entire manuscript to make sure that when referring to the gestational profile, we specify that miR-210-3p levels were higher in the first trimester than in the third trimester. In addition, we did some minor language editing.